# Effects of Ascorbic Acid on Physiological Characteristics during Somatic Embryogenesis of *Fraxinus mandshurica*

**DOI:** 10.3390/ijms24010289

**Published:** 2022-12-24

**Authors:** Xue Cheng, Tianyi Xie, Ling Yang, Hailong Shen

**Affiliations:** 1State Key Laboratory of Tree Genetics and Breeding, School of Forestry, Northeast Forestry University, Harbin 150040, China; 2State Forestry and Grassland Administration Engineering Technology Research Center of Native Tree Species in Northeast China, Harbin 150040, China; 3State Forestry and Grassland Administration Engineering Technology Research Center of Korean Pine, Harbin 150040, China

**Keywords:** *Fraxinus mandshurica*, somatic embryogenesis, ascorbic acid, embryo explants, explant browning

## Abstract

*Fraxinus mandshurica* is one of the precious tree species in northeast China and has important economic and ecological value. Ascorbic acid (ASA) is a strong antioxidant that can significantly improve plant photosynthetic efficiency and stress resistance and participate widely in plant growth and development. In this study, we investigated the development process of mature zygotic embryos of *F. mandshurica* under different concentrations of ASA and found that 100 mg·L^−1^ exogenous ASA was the optimal concentration and that the induction rate of somatic embryos (SEs) was the highest at 72.89%, which was 7.13 times higher than that of the control group. The polyphenol content, peroxidase (POD) activity, nitric oxide (NO) content, nitrate reductase (NR) activity, total ascorbic acid (T-ASA) content, ASA content, ASA/Dehydroascorbic acid (DHA) ratio, GSH/GSSG ratio, and ascorbate peroxidase (APX) activity were significantly increased under the application of exogenous ASA in explants, whereas the polyphenol oxidase (PPO) activity, phenylalanine ammonia-lyase (PAL) activity, superoxide dismutase (SOD) activity, and catalase (CAT) activity, malondialdehyde (MDA) content and nitric oxide synthase (NOS) activity were decreased. At the same time, the content of T-ASA and ASA, T-GSH and GSSG, and PAL and SOD had the same change pattern in the control group and the treatment group. These results suggested that high or low concentrations of ASA could not promote the somatic embryogenesis of *F. mandshurica* and that exogenous ASA had significant effects on the physiology of *F. mandshurica* explants. ASA was also highly related to somatic embryogenesis and the explant browning of *F. mandshurica.* Our results could provide a reference for further study on the browning mechanism of *F. mandshurica* explants and lay the foundation for optimizing the condition of somatic embryogenesis in *F. mandshurica.*

## 1. Introduction

Somatic embryogenesis is the production of somatic embryos (SEs) from vegetative cells as well as the induction of totipotency in the produced SEs to form plants in vitro [1]. It is an effective system for the accelerated propagation of plants as well as a good model system for research into plant embryogenesis, somatic cell differentiation and development, and histological and physiological characteristics [2,3,4,5].

*Fraxinus mandshurica* Rupr. (*Manchurian ash*) was an economically and ecologically important tree species with high quality and beautiful timber and an important community position in the temperate mixed forest of Korean pine and broadleaved tree species in northeast Asia. Due to the wide availability of *F. mandshurica*, unreasonable logging and utilization, and the need for long-term stratification of seeds to break dormancy, its production and practical application faced serious obstacles. SEs technology can greatly improve the reproduction coefficient and is an effective means to achieve propagation of *F. mandshurica*, and its SEs have been obtained by our team [6,7,8]. We found that most of the SEs (79–92%) of *F. mandshurica* were generated on browned explants [8,9] and this phenomenon could also be found in other species like Eucalyptus [10], and shown on SE regeneration figures in various papers. Beneficial explant browning might be caused by not only polyphenols but programmed cell death as well [11]. In the process of studying the reasons and regulation methods for the explant browning of *Manchurian ash* [9,11,12,13], we found that ascorbic acid has promotion effects on explant browning as well as SE induction. 

Ascorbic acid (ASA) is a kind of small molecular antioxidant in plants that plays an important role in plant resistance to oxidative stress. Furthermore, as a cofactor of enzymes related to cell metabolism [14,15], ascorbic acid is widely involved in regulating plant growth and physiological function. A previous study found that exogenous ASA treatment increased the seed germination rate and salt tolerance of alfalfa seedlings [16]. Similarly, spraying ASA on the heads of *Calendula officinalis* can help plants resist salt stress and increase the stability of proline and cell membranes [17]. Exogenous ASA can reduce the degree of plasma membrane oxidation in rice, improve the activity of antioxidant enzymes, increase the content of osmotic adjustment substances and antioxidant substances, increase the disease resistance of rice, and reduce the accumulation of H_2_O_2_ [18]. These results indicated that exogenous ASA plays a key role in regulating the balance of reactive oxygen species, which can effectively improve plant stress resistance and reduce oxidative damage. However, the effect of ASA on the physiology of different stages of somatic embryogenesis in *F. mandshurica* is still unclear.

Based on an analysis of the developmental process of somatic embryogenesis at different stages, the effect of exogenous ASA on somatic embryogenesis in mature zygotic embryos of *F. mandshurica* treated with different concentrations of ASA was studied. It will provide a theoretical basis for the application of ASA to the high frequency of somatic embryogenesis and contribute to understanding the mechanism of browning during somatic embryogenesis in *F. mandshurica*.

## 2. Results

### 2.1. Effects of Exogenous ASA on Somatic Embryogenesis and Explant Browning of F. mandshurica

Exogenous ASA had significant effects on somatic embryogenesis and the browning of mature zygotic embryo explants of *F. mandshurica* (Figure 1).

In the process of somatic embryo induction culture of *F. mandshurica*, the initial cotyledons were milky white (A1, A2), and the cotyledons gradually became larger and irregularly curved after four days of inoculation. At seven days, most of the inoculated cotyledons were curved (B1, B2). With the growth of explants, brown spots gradually appeared on their surfaces. The time of brown spots in the control group (8 d) was earlier than that in the 100 mg·L^−1^ ASA treatment group (10 d) (C1, C2). In the later stage, the brown spots on the explants increased, most of which spread throughout the whole explant (D1, D2). Translucent globular embryos were observed in the 100 mg·L^−1^ ASA treatment group after 23 days, and somatic embryos gradually appeared in the control group after 30 days (E1, E2). With the increase in culture time, the number of somatic embryos gradually increased and gradually turned into milky white cotyledon embryos, and the number of somatic embryos in the control group was significantly lower than that in the 100 mg·L^−1^ ASA treatment group (F1, F2) (Figure 1A).

With the increase in ASA concentration, the somatic embryo induction (SEI) rate and explant browning (EB) rate showed a down-up trend. However, when the concentration of ASA was 150 mg·L^−1^, all of the inoculated cotyledon explants of *F. mandshurica* were dead (there was no change in the cotyledons at 56 d, which was identified as a dead state, which is not shown in Figure 1B), whereas, when the concentration of ASA was 100 mg·L^−1^, the SEI rate was the highest (72.89%), which was 7.13 times that of the control group. The EB rate reached 98% in treatments with the 50 mg·L^−1^ and 100 mg·L^−1^ concentrations of ASA, the same as that of the control group, whereas, those of the remaining treatment groups were lower than those of the control group (Figure 1B).

### 2.2. Effect of 100 mg·L^−1^ ASA on Somatic Embryogenesis of Heterogenous F. mandshurica

According to the above results, the highest SEI rate was identified under 100 mg·L^−1^ ASA. However, the effect of 100 mg·L^−1^ ASA on materials from different provenances of *F. mandshurica* was still unclear. The statistical test indicated that there were significant differences in SEI rates between the control group and treatment group of *F. mandshurica* from different provenances. The changing trend between the control group and the treatment group was the opposite. In the control groups, the SEI rate of provenance 3 was the highest, at 30.67%, while the SEI rate of provenance 1 was the lowest, at 10.22%. Under 100 mg·L^−1^ ASA, the SEI rate of provenance 1 was 7.13 times that of the control group. Particularly, the SEI rate in provenance 3 was 62.85% and higher than that in the control group. Combined with practical comprehensive analysis, provenance 3 was used for later physiological and biochemical analysis (Figure 2).

### 2.3. Effects of Exogenous ASA on Browning of Explants during Somatic Embryogenesis of F. mandshurica

The physiological analysis found that exogenous ASA had a significant effect on the polyphenol content during the somatic embryogenesis of *F. mandshurica*. All of the control and treatment groups showed a downward, ascending, descending, and ascending trend. Furthermore, exogenous ASA significantly increased the polyphenol content of *F. mandshurica* explants. The content of polyphenols in the control and treatment groups was the highest on the 14th day, which was 0.56 mg·L^−1^ FW, and 0.69 mg·L^−1^ FW, while the lowest was on the 7th day, which was 0.23 mg·L^−1^ FW and 0.56 mg·L^−1^ FW, respectively (Figure 3a). Particularly, the PPO activity of 5–7 d and 35–42 d was higher than that of the control group, while the PPO activity of the remaining groups was lower than that of the control group. It increased at first and then decreased during the early and middle stages (5–42 d). In the later stage (42 d), the PPO activity in the control group increased gradually, while it significantly decreased in the treatment groups. The PPO activity of the control group and treatment group reached its peak on 14 d, which was 136.74 μ·g^−1^ FWmin^−1^ and 129.17 μ·g^−1^ FW min^−1^, respectively (Figure 3b). 

The PAL activity of the treatments was higher than that of the control group on the 49 d, and the other activities were lower than that of the control group. In the early stage (5–7 d), the control and treatment groups showed the opposite trend, and they showed an upward and downward trend in the middle and later stages, respectively. PAL activity on 14 days was consistent with polyphenol content and PPO activity with a significant peak. The PAL activity of the control group and treatment group was 30,832.2 μ∙g^−1^ FW∙h^−1^ and 20,131.1 μ∙g^−1^ FW∙h^−1^, respectively (Figure 3c).

During the somatic embryogenesis of *F. mandshurica*, the polyphenols, PPO activity, and PAL activity in the control group and the treatment group were the same and reached a peak at 14 d. The content of polyphenols obviously increased, and PPO activity and PAL activity decreased using exogenous ASA. In the peak, the polyphenols content of the treatment group was higher than that in the control group, and the activity of PPO and PAL displayed the opposite trend.

### 2.4. Effects of Exogenous ASA on Activities of Antioxidant Enzymes during Somatic Embryogenesis of F. mandshurica 

The POD activity obviously increased under exogenous ASA treatment during the somatic embryogenesis of *F. mandshurica*. The trend of SOD activity in CK and 100 mg·L^−1^ ASA was opposite in the early stage (5–7 d) and showed an upward-downward trend in the middle and later stages. An obvious peak was found in 14 d, and the SOD activities of the control group and treatment group were 1229.79 μ∙g^−1^ FW and 652.84 μ∙g^−1^ FW, respectively (Figure 4a). The POD activity of CK and the treatment group reached a peak on 56 d, which were 1116.4 μ∙g^−1^ FWmin^−1^ and 1153.2 μ∙g^−1^ FW min^−1^, respectively (Figure 4b).

The CAT activity in the treatment group (100 mg·L^−1^ ASA) was slightly higher than that of the control group on 28 d. The trend was as follows: first increased and then decreased, reached the peak on the 21st day, and the CAT activity was 947.3 μ·g^−1^ FW min^−1^ and 878.2 μ·g^−1^ FW min^−1^, respectively (Figure 4c).

In addition, the SOD activity and CAT activity of explants decreased under 100 mg·L^−1^ ASA, while the POD activity increased during somatic embryogenesis of *F. mandshurica*, and they maintained the same change trend in CK and the treatment group (except 5–7 d of SOD). The peaks of SOD, CAT activity, and POD activity appeared on 14 d, 21 d, and 56 d, respectively.

### 2.5. Effects of Exogenous ASA on Intracellular H_2_O_2_ during Somatic Embryogenesis of F. mandshurica

Exogenous ASA had a significant effect on intracellular H_2_O_2_ during the somatic embryogenesis of *F. mandshurica*. The content of H_2_O_2_ in explants increased in the early stage (5–21 d), while it decreased gradually in the later stage. The highest peak value was found on the 21 d in CK and the treatment group, and the content of H_2_O_2_ in the treatment group was 18.93% higher than that in CK. Particularly, the lowest H_2_O_2_ content was at 56 d (Figure 5).

### 2.6. Effects of Exogenous ASA on Intracellular MDA in Explants during Somatic Embryogenesis of F. mandshurica

The content of MDA in the treatment group with exogenous ASA was higher than that in the control group on the 5 d and 21 d, and the MDA content of the remaining treatment group was lower than that in the control group. Particularly, the content of MDA in the control and treatment groups increased at first and then decreased. During the development of somatic embryos, the highest MDA content in the control group and the treatment group was found at 7 d (Figure 6).

### 2.7. Effects of Exogenous ASA on Intracellular NO Synthesis and Metabolism during Somatic Embryogenesis of F. mandshurica

The changing trend of intracellular NO content of *F. mandshurica* explants in the control group was the same as that in the treatment group: down-up-down-up-down. During somatic embryogenesis of *F. mandshurica*, the NO content in the treatment group was lower than that of the control group on the 28 d and 42 d. The highest intracellular NO content of CK and the treatment group was found at 5 d, in which NO content under 100 mg·L^−1^ ASA was significantly higher than that of the control group with 55.61% (Figure 7a).

The intracellular NOS activity decreased in the early stage (5–21 d) under 100 mg·L^−1^ ASA treatment, while the intracellular NOS activity during the middle and later stages in the treatment group was higher than that in the control group. On the 7th day, the intracellular NOS activity of the control group was the highest with 0.64 U/mgprot, which was 50.49% higher than that of the treatment group. On the 28th day, the intracellular NOS activity of the treatment group was the highest with 0.54 U/mgprot, which was 4.6 times higher than that of the control group (Figure 7b).

The activity of intracellular NR in the early stage (7–21 d) under 100 mg·L^−1^ ASA was significantly increased, and the activity of intracellular NR in the later treatment group was lower than that in the control group. The intracellular NR activity of the treatment group was the highest on the 14th day, which was 88.42% higher than that of the control group. In addition, the intracellular NR activity in the control group was the highest on the 56th day, which was 50% higher than that of the treatment group. Particularly, the activity of NR was the lowest on the 5th day in both the control group and the treatment group (Figure 7c).

NO content and NR activity of the explant increased during the early stage (5–21 d) under 100 mg·L^−1^ ASA, and the NOS activity decreased. On the 42nd day, the NO content, NR activity, and NOS activity in the treatment group were lower than that in the control group. From 49 to 56 d, the NO content and NOS activity in the treatment group were also higher than that in the control group.

### 2.8. Effects of Exogenous ASA on ASA and GSH Content during Somatic Embryogenesis of F. mandshurica

The content of T-ASA under 100 mg·L^−1^ ASA increased during the embryogenesis of *F. mandshurica* somatic. The changing trend of T-ASA content in the control and treatment groups was the same: up-down-up-down. The highest peak in the control group was found on the 42nd day, which was 30.75% lower than that of the treatment group. The content of T-ASA in the treatment group was the highest on the 7th day, which was 2.1 times of the control group. The content of T-ASA in the control group and the treatment group was the lowest on the 5th day. (Figure 8a).

In addition, endogenous ASA content also increased under exogenous ASA. The changing trend of the control group and treatment group was: ascending and then descending. The ASA content was the lowest on the 5th day in the control and treatment groups. Furthermore, the ASA content of the control group was the highest on the 42nd day, while the ASA content of the treatment group was the highest on the 7th day (Figure 8b).

In the early stage (5–14 d), the intracellular DHA content of the control group was higher than that of the treatment group, but it was lower than that of the treatment group in other stages. The changing trend of the control and treatment groups was the same, rising first and then descending. The peak value of the control group was found on the 14th day, and it was 0.47 μ mol·g^−1^ FW. The peak of the treatment group was on the 21st day, which was 4.5 times higher than that of the control group. On the 35th day, the content of DHA in both the control group and the treatment group was the lowest (Figure 8c).

The ratio of ASA/DHA increased under exogenous ASA, and the changing trend of ASA/DHA was the same in the control and treatment groups. The peak of the control group was found on the 7th day, while the peak of the treatment group was found on the 42nd day. (Figure 8d).

Exogenous ASA increased the T-ASA content, ASA content, and ASA/DHA ratio of *F. mandshurica*, which inhibited DHA content in the early stage (7–14 d) and promoted it in the later stage. Exogenous ASA had an effect on T-ASA content and ASA content in vitro of *F. mandshurica* in the early stage (7 d), the ratio of ASA/DHA in the later stage (42 d), and the content of DHA in the middle stage (21 d), which had a significant promoting effect compared with the control group.

### 2.9. Effects of Exogenous ASA on GSH Content during Somatic Embryogenesis of F. mandshurica

During the somatic embryogenesis of *F. mandshurica*, the changing trend of T-GSH content in the control group at 14–42 d was the same as in the treatment group. At 5–7 d, 28 d, and 56 d, the T-GSH content in the treatment group was higher than that in the control group. The T-GSH content in the control group was the highest on the 42nd day, which was 1.95 times higher than that in the treatment group. The T-GSH content was the lowest on the 28th day, while the highest was on the 7th day in the treatment group, which was 3.68 times that of the control group (Figure 9a).

The GSH content in the treatment group was higher than that in the control group at 5–14 d and 35 d. The changing trend of the treatment group was the same as that of the control group: ascending-descending mode. The peak of GSH content in the control group was on the 49th day, which was 2.1 times higher than that of the treatment group, and the lowest peak value was on the 35th day. The GSH content in the treatment group was the highest on the 14th day, which was 3.85 times higher than that of the control group, and the lowest peak value was on the 42nd day (Figure 9b).

The changing trend of GSSG content under 100 mg·L^−1^ ASA was the same as that of T-GSH. During 14–42 d, the changing pattern of the treatment group and control group was the same. The GSSG content in the control group first reached the lowest point and then increased, while in the treatment group it was the opposite (Figure 9c).

The ratio of GSH/GSSG under 100 mg·L^−1^ ASA was lower than that of the control group on the 5th day. The changing trend of the control and treatment groups was the same over the 5–42 d period: descending—ascending—descending—ascending. However, during 42–56 d, the changing trend of the GSH/GSSG ratio in the control group and the treatment group was the opposite. The lowest values in the control and treatment groups appeared on the 28th day (Figure 9d).

The changing trend of T-GSH content under 100 mg·L^−1^ ASA was the same as that of GSSG content. Particularly, the GSH content under 100 mg·L^−1^ ASA increased in the early stage (5–14 d) and decreased in the later stage. In addition to 5 d, the GSH/GSSG ratio under 100 mg·L^−1^ ASA was higher than that of the control group. 

### 2.10. Effects of Exogenous ASA on APX Activity in Explants during Somatic Embryogenesis of F. mandshurica

Under 100 mg·L^−1^ ASA, intracellular APX activity significantly increased, and the 5–7 d changing trend was the opposite, and the changing trend was the same in the later stage. The APX activity of the treatment group was the highest on the 5th day, which was 5.24 times higher than that of the control group. The APX activity of the control group was the highest on the 28th day. The lowest APX activity of the control and treatment groups was found on the 14th day (Figure 10).

## 3. Discussion

### 3.1. Exogenous ASA Could Synchronously Promote Explant Browning and Somatic Embryogenesis of F. mandshurica 

ASA is often used as an antioxidant to reduce the browning of explants and ensure normal plant growth [11,12,13,19]. In the present study, the results showed that the changing trends of the SEI rate and EB rate of *F. mandshurica* were in the same pattern: the EB rate and SEI rate were in the same trend under low concentrations of ASA, while they were in the opposite pattern under higher concentrations of ASA. This is contrary to the basic effect of antioxidants [12,19] but has been confirmed by our previous studies [11,12,13] and the results were consistent with those of fresh-cut water *Eleocharis tuberosa* treated with exogenous ASA in that it failed to inhibit its yellowing [20]. The browning of explants can be divided into enzymatic browning and non-enzymatic browning, and the former is the main one. Polyphenols, PPO, and PAL are the main enzymes that cause explant browning [21]. PPO is abundant in plant cells and can use oxygen to oxidize phenols to quinones. While under external stimulation, PAL activity accelerates the decomposition of phenylalanine, resulting in the synthesis of phenols, which leads to explant browning under the combination of phenols and oxygen [22]. In this study, it was found that the changes in polyphenols, PPO activity, and PAL activity in the control group during somatic embryogenesis of *F. mandshurica* were the same as that of the treatment group. The peak value of them was found on the 14th day, indicating that the peak period of phenolic compounds during the somatic embryogenesis of *F. mandshurica* appears on the 14th day. Exogenous ASA promoted polyphenol content in the whole process, promoted PPO and PAL activities in a short time in the later period, and inhibited PPO and PAL activities in the rest. This is different from the result of inhibiting PPO content but not PAL content in fresh-cut water chestnuts treated with exogenous ASA [20]. The results of the browning test of fresh-cut Sagittaria (*Sagittaria sagittifolia*) showed that the total phenol content, PPO activity, and PAL activity decreased under exogenous ascorbic acid [23]. These results showed that different explants displayed distinct internal physiologic changes under exogenous ASA, and may be the available cues for understanding their function during somatic embryogenesis of *F. mandshurica*.

### 3.2. Exogenous ASA Showed Diverse Patterns on Antioxidant Enzymes in Explants of F. mandshurica

SOD, POD, and CAT are protective enzymes with the ability to scavenge active oxygen. Particularly, SOD acts to remove superoxide anions and generate H_2_O_2_, and POD catalyzes H_2_O_2_, phenolic, and amine compounds, CAT is responsible for scavenging peroxides produced during photorespiration or fatty acid oxidation [24]. In this study, it was found that the SOD and CAT activities of explants decreased under exogenous ASA during somatic embryogenesis of *F. mandshurica*, but the POD activity increased, which was opposite to the results of *Dimocarpus longan* under exogenous ASA [25]. The peak of SOD activity appeared in the early stage (14 d), the peak of CAT activity was found in the middle stage (21 d), and the peak of POD activity appeared in the later stage (56 d), which indicated that the ability to scavenge active oxygen increased under exogenous ASA in the early and middle stages, but decreased in the later stage during the somatic embryogenesis of *F. mandshurica*. We speculated that the heterogeneity of explants may be the main reason for the changes in antioxidant enzymes under exogenous ascorbic acid.

### 3.3. Exogenous ASA Has Regulation Function for Reducing Oxidative Damage in Explants of F. mandshurica

H_2_O_2_ is an important plant oxidative metabolism and has strong oxidation, and it can also form reactive oxygen species (ROS) with singlet oxygen (^1^O_2_), superoxide anion (O_2_·^−^), and hydroxyl radical (OH) [26]. MDA is one of the decomposition products of cell membrane lipid peroxidation, and its content represents the degree of cell membrane peroxidation and damage [27]. In the present study, the results showed that the H_2_O_2_ content of explants increased in the early stage (5–21 d) under exogenous ASA but decreased in the later stage, which was consistent with the results of browning of explants and somatic embryos in *F. mandshurica*. Under exogenous ASA, the MDA and H_2_O_2_ contents first increased and then decreased. The MDA content was the highest on the 7th day, while the H_2_O_2_ content was the highest on the 21st day. These results suggested that the early release of H_2_O_2_ and MDA caused a high level of lipid peroxidation in the inner membrane and promoted the browning of *F. mandshurica* explants, while in the later stage, the H_2_O_2_ and MDA content decreased, which accelerated the clearance of intracellular ROS and increased the somatic embryogenesis of *F. mandshurica*. The results above were consistent with the results that H_2_O_2_ and MDA content decreased in *Oryza sativa* under exogenous ASA treatment, alleviated salt damage, and enhanced salt tolerance in rice [28]. It is suggested that ASA has a protective regulation function for reducing oxidative damage of explants.

### 3.4. Exogenous ASA Increase NO Content and NR Activity of F. mandshurica Explants

NO plays an important role as a signal molecule in plant growth and development in response to biotic and abiotic stress, fruit ripening, flowering regulation, etc. In general, NO synthesis in plants can be divided into non-enzymatic reactions and enzymatic reactions. NOS and NR in the enzymatic reaction are the main pathways of NO synthesis [29]. In this study, the results showed that NO content and NR activity increased in vitro under exogenous ASA, and intracellular NOS activity decreased in the early stage (5–21 d), which was the opposite in the later stage. The results showed that NO content and NR activity played a leading role in the early stage of exogenous ASA treatment, and NOS activity played a leading role in the later stage of exogenous ASA treatment. These results were opposite to that of [30], which may be the result of different exogenous additives. Particularly, NOS and NR activity coordinately regulate the synthesis of NO, NO content, and NR activity may be related to the browning of *F. mandshurica* explants, while NOS activity may be involved in the somatic embryogenesis of *F. mandshurica*.

### 3.5. Exogenous ASA on ASA and GSH Content and APX Activity in F. mandshurica Explants

ASA and GSH are non-enzymatic antioxidants for plants scavenging H_2_O_2_, and they can react with ROS directly to reduce ROS [31]. In the ASA-GSH cycle, GSH is oxidized to GSSG during scavenging reactive oxygen, and the GSH/GSSG ratio is an important sign to measure the redox potential of GSH. Similarly, ASA/DHA is also a crucial index to reflect the redox state of intracellular ASA [32]. APX is a peroxidase with ASA as an electron donor, which is the main enzyme for scavenging H_2_O_2_ [33]. In this study, T-ASA content, ASA content, and the ASA/DHA ratio increased under exogenous ASA, while DHA content in the early stage (7–14 d) decreased in *F. mandshurica*. The peaks of T-ASA and ASA content were found on the 7th day, the peak of DHA was on the 21st day, and the peak of ASA/DHA was on the 42nd day, indicating that the somatic embryogenesis of the early stage, middle stage, and later stage of *F. mandshurica* maintains reduction, oxidation, and reduction state under exogenous ASA treatment. Furthermore, the T-GSH and GSSG showed a similar trend under exogenous ASA: they increased at 5–7 d, 28 d, and 56 d, but decreased at other periods. Regarding GSH, it increased at 5–14 d and 35 d and decreased at other times. Under ASA treatment, the reducibility of explants significantly improved in addition to the 5th day. At the same time, the contents of T-ASA and ASA, T-GSH, and GSSG had the same change pattern in the control group and the treatment group. The activity of APX increased during the somatic embryogenesis of *F. mandshurica*. The peak of APX was found at the initial stage, and the APX activity was the lowest on the 14th day. The APX activity showed different change patterns in different stages, and the specific reasons need to be confirmed. These results were consistent with the results of apple (*Malus domestica*) under exogenous ASA, which contribute to promoting APX, ASA, GSH activity, and ASA/DHA [34].

## 4. Materials and Methods

### 4.1. Plant Materials

Somatic embryogenesis was performed in this study using mature *F. mandshurica* seeds (seed wing color brown, dehydrated) from four different provenance mother trees. The provenance information is as follows: The mixed mature seed of tree No. 1 was collected from Maoershan Experimental Forest Farm of Northeast Forestry University, Heilongjiang Province, China (127°18′0″ E, 45°2′20″ N) in December 2016; the mixed mature seed of tree No. 2 was obtained from Linjiang Seed Orchard of Jilin Province, China (127°18′0″ E, 45°2′20″ N) in December 2016; the mixed mature seed of tree No. 3 was taken from the campus of Northeast Forestry University in Harbin City in Heilongjiang Province, China (127°18′0″ E, 45°2′20″ N) in September 2018; the mixed mature seed of tree No. 4 was sampled from Qingshan Forest Farm in Weihe Forest Enterprise in Heilongjiang Province, China (127°18′0″ E, 45°2′20″ N) in September 2018. All of the sampled experimental materials were stored in the refrigerator at 4 °C after harvest.

### 4.2. Preparation of Explants of F. mandshurica

To obtain the sterile explant materials, the collected samples were further sterilized. Firstly, the seed wings were stripped, then the seeds were placed in a beaker, and washed with running water for three days. Secondly, the washed seeds were sterilized on the surface of the clean bench. 75% alcohol disinfected for 30 s, 5% sodium hypochlorite for 15 min, then rinsed four to six times with sterile water then set aside. Finally, during inoculation, 1/3 of the seed hypocotyls were removed with a sterile dissector. The seed embryo was extruded, and the single cotyledon (inside cotyledon) was inoculated onto the induction medium.

### 4.3. Induction of Somatic Embryogenesis

The following medium was used to induce SE in *F. mandshurica*: MS1/2 + 400 mg·L^−1^ acid hydrolyzed casein (CH) + 75 g·L^−1^ sucrose + 0.5 mg·L^−1^ 6-BA + 1.5 mg·L^−1^ NAA + 7.5 g·L^−1^ agar, and the pH value was adjusted to 5.8. The medium was sterilized at 121 °C for 20 min under high temperature and high pressure. In addition, exogenous ASA (filter sterilizer, aperture 0.22 μm) should be added after sterilization of the medium. The concentrations of ASA were 10 mg·L^−1^, 20 mg·L^−1^, 50 mg·L^−1^, 100 mg·L^−1^, and 150 mg·L^−1^, each treatment consisted of five culture plates, and there were 10 explants in each culture plate. The explants were all grown under the same conditions: 24 ± 2 °C, 40–70% humidity, and dark culture. Update the same medium every 30 days. From the day of inoculation, the phenotypic changes of the explants were observed under a stereomicroscope (SZX-ILLB2-200, Tokyo, Japan) (Figure 1A).

### 4.4. Determination Indicators and Methods

Samples were taken at 5 d, 7 d, 14 d, 21 d, 28 d, 35 d, 42 d, 49 d, and 56 d (0.3 g per treatment, 3 replicates, the sampling material was inoculated explants containing late growth embryos), and immediately stored at −80 °C for subsequent physiological analysis. Particularly, polyphenol content, PPO activity, POD activity, CAT activity, and SOD activity were determined according to the methods described by [35]. Furthermore, the measure of MDA content and PAL activity was described in [36]. H_2_O_2_ and NO contents were determined based on the method described by [37]. NOS and NR enzyme activity was determined according to [38]. The content of ASA and DHA was determined by the Fe^3+^ reduction method described by [39]. The content of GSH and GSSG was determined based on the method described by [40], and the activity of APX was determined based on the method described by [41].

### 4.5. Statistical Analysis of Data

The descriptive statistics of all collected data were performed by Microsoft Excel (2019) and by Sigmaplot (v14.0, SYSTAT USA) software to draw the graph. The statistical test and multiple comparisons were obtained by SPSS (v21.0, SPSS Inc. USA) software. Error bars show the standard error of the mean. At 56 days of induction culture, the number of browning explants and somatic embryos were counted, and the browning rate of explants and the incidence of somatic embryos were calculated.

The calculation formula is as follows: SEI (somatic embryo induction rate) rate (%) = number of explants producing somatic embryos/number of explants survived by inoculation × 100, and EB (explants browning) rate (%) = number of browning explants/numbers of living explants inoculated × 100.

## 5. Conclusions

The 100 mg·L^−1^ ASA treatment could significantly increase the SE induction rate of *F. mandshurica*, and the changes in SE induction rate and explant browning rate were synchronous. The somatic embryo induction rate of explants from different provenances was different. Under exogenous ASA treatment, the polyphenol content, POD activity, NO content, NR activity, T-ASA content, ASA content, ASA/DHA ratio, GSH/GSSG ratio, and APX activity in explants were increased, but PPO activity, PAL activity, SOD activity, CAT activity, MDA content, and NOS activity were decreased. In addition, exogenous ASA treatment also increased the H_2_O_2_ content in explants in the early stage (5–21 days of culture), and decreased the content of H_2_O_2_ in the later stage. The DHA content decreased in the early stage (7–14 days of culture) but increased in the later stage under exogenous ASA. Particularly, Exogenous ASA promoted the content of T-GSH, GSH, and GSSG in the early stage (5–7 d), while ASA inhibited their content in the middle and late stages. At the same time, the content of T-ASA and ASA, T-GSH and GSSG, and PAL and SOD, had the same change pattern in the control group and the treatment group. In summary, this study revealed the effect of antioxidant ASA on explant browning and somatic embryogenesis of *F. mandshuric* and found that it indirectly changed the frequency of somatic embryogenesis by changing the internal physiological activity. Subsequently, the mechanism of action of antioxidants on somatic embryogenesis can be further analyzed through the whole genome, providing theoretical support for the antioxidant regulation of the somatic embryogenesis system.

## Figures and Tables

**Figure 1 ijms-24-00289-f001:**
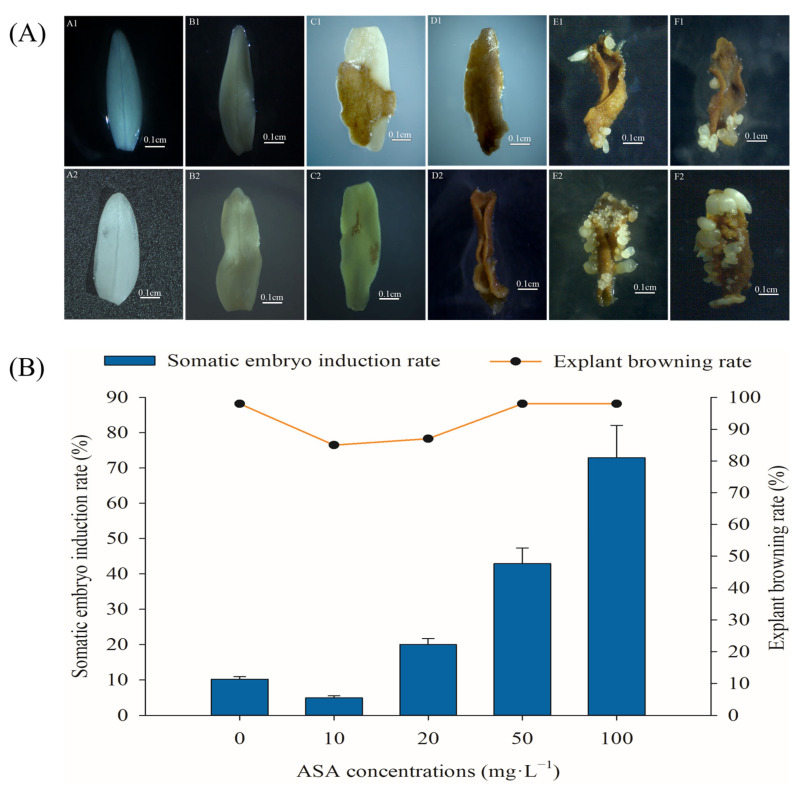
Morphological changes during somatic embryogenesis of *Fraxinus mandshurica* explants (**A**). Changes in different developmental stages of somatic embryogenesis in the control group (**A1**–**F1**). Changes in different developmental stages during somatic embryogenesis in the 100 mg·L^−1^ ASA treatment group (**A2**–**F2**). State of explants at 0 d (**A1**, **A2**), 7 d (**B1**, **B2**), 14 d (**C1**, **C2**), 21 d (**D1**, **D2**), 42 d (**E1**, **E2**), 56 d (**F1**, **F2**). Effects of different concentrations of ascorbic acid on somatic embryogenesis and the browning of mature zygotic embryo explants of *F. mandshurica* provenance 1 (**B**). The *x*-axis represents the ASA concentration, and the *y*-axis on the left represents the somatic embryo induction rate, the *y*-axis on the right represents the explant browning rate. Error bars show the standard error of the mean.

**Figure 2 ijms-24-00289-f002:**
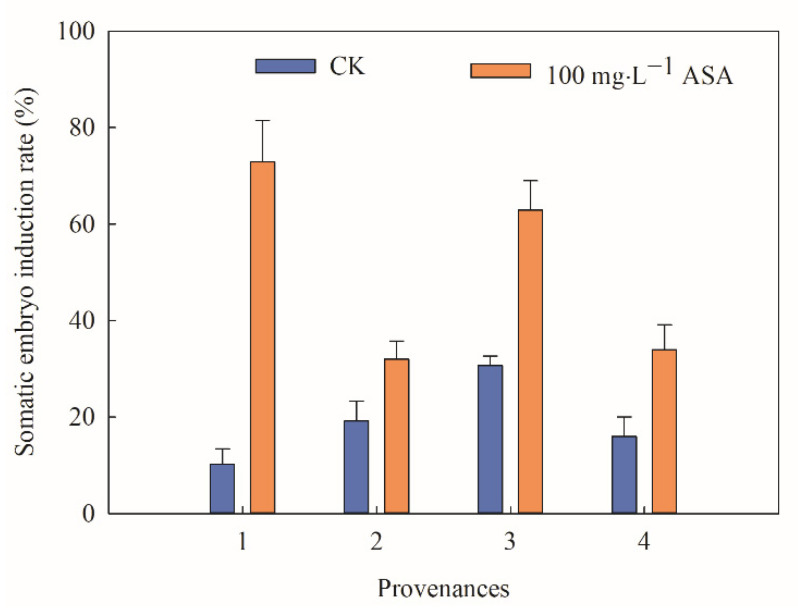
Effects of 0 and 100 mg·L^−1^ ASA treatments on somatic embryogenesis of *Fraxinus mandshurica* from different provenances. The *x*-axis represents the different provenances of *F. mandshurica*, and the *y*-axis represents the somatic embryo induction rate (%). Error bars show the standard error of the mean.

**Figure 3 ijms-24-00289-f003:**
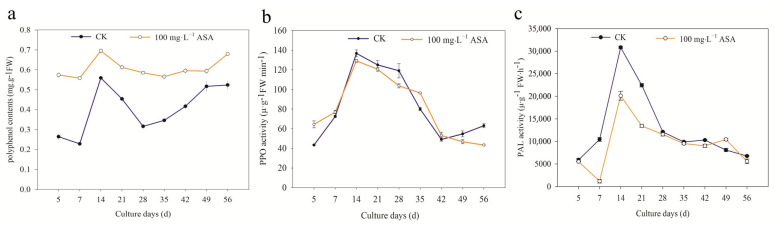
Effects of 0 and 100 mg·L^−1^ ASA treatments on polyphenols, PPO activity, and PAL activity during somatic embryogenesis of *Fraxinus mandshurica* provenance 3. The *x*-axis represents the culturing days, and the *y*-axis represents the intracellular polyphenol content (**a**), intracellular PPO activity (**b**), and intracellular PAL activity (**c**). Error bars show the standard error of the mean.

**Figure 4 ijms-24-00289-f004:**
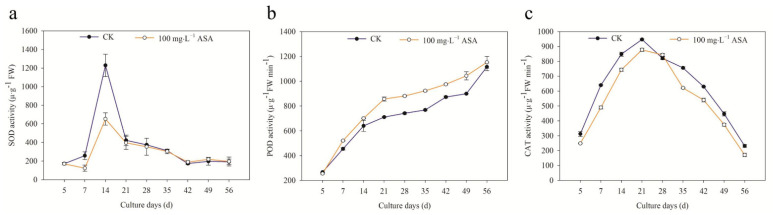
Effects of 0 and 100 mg·L^−1^ ASA treatments on SOD activity, POD activity, and CAT activity during somatic embryogenesis of *Fraxinus mandshurica* provenance 3. The *x*-axis represents the culturing days, and the *y*-axis represents the SOD activity (**a**), POD activity (**b**), and CAT activity (**c**). Error bars show the standard error of the mean.

**Figure 5 ijms-24-00289-f005:**
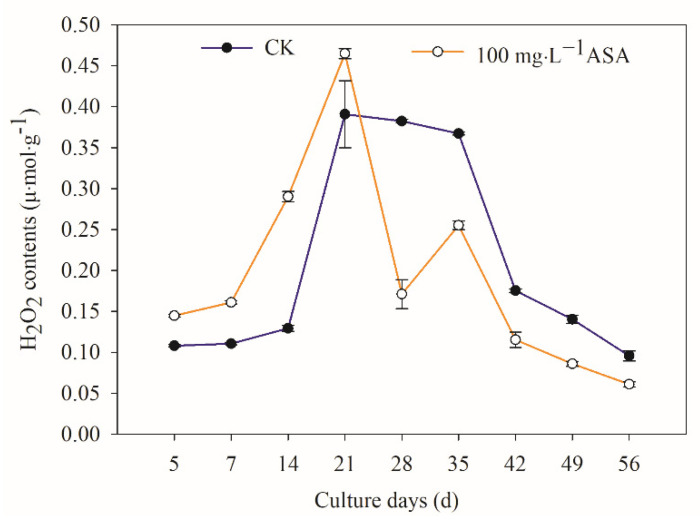
Effects of 0 and 100 mg·L^−1^ ASA treatments on intracellular H_2_O_2_ content during somatic embryogenesis of *Fraxinus mandshurica* provenance 3. The *x*-axis represents the culturing days, and the *y*-axis represents the intracellular H_2_O_2_ content. Error bars show the standard error of the mean.

**Figure 6 ijms-24-00289-f006:**
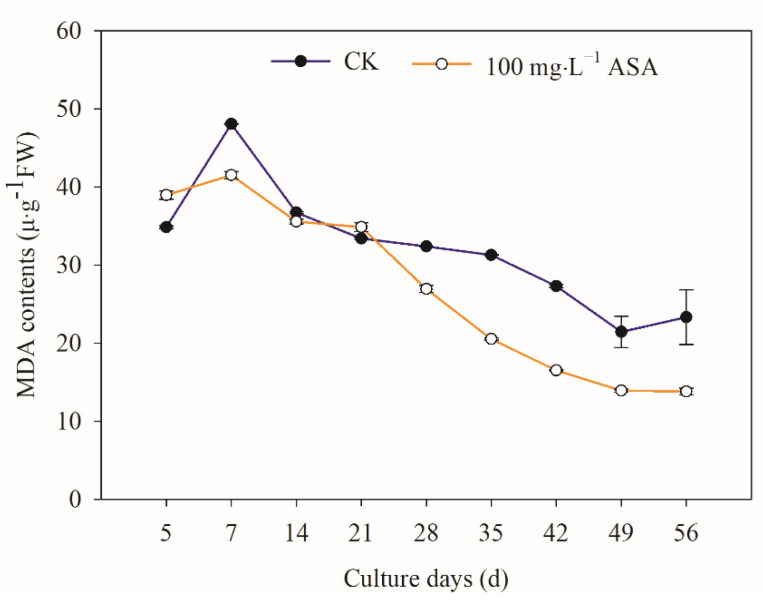
Effects of 0 and 100 mg·L^−1^ ASA treatments on intracellular MDA content during somatic embryogenesis of *Fraxinus mandshurica* provenance 3. The *x*-axis represents the culturing days, and the *y*-axis represents the intracellular MDA content. Error bars show the standard error of the mean.

**Figure 7 ijms-24-00289-f007:**
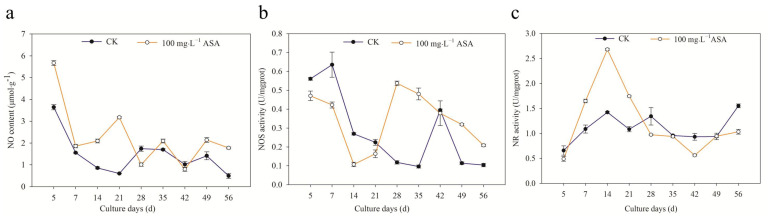
Effects of 0 and 100 mg·L^−1^ ASA treatments on NO content, NOS activity, and NR activity during somatic embryogenesis of *Fraxinus mandshurica* provenance 3. The *x*-axis represents the culturing days, and the *y*-axis represents the NO content (**a**), NOS activity (**b**), and NR activity (**c**). Error bars show the standard error of the mean.

**Figure 8 ijms-24-00289-f008:**
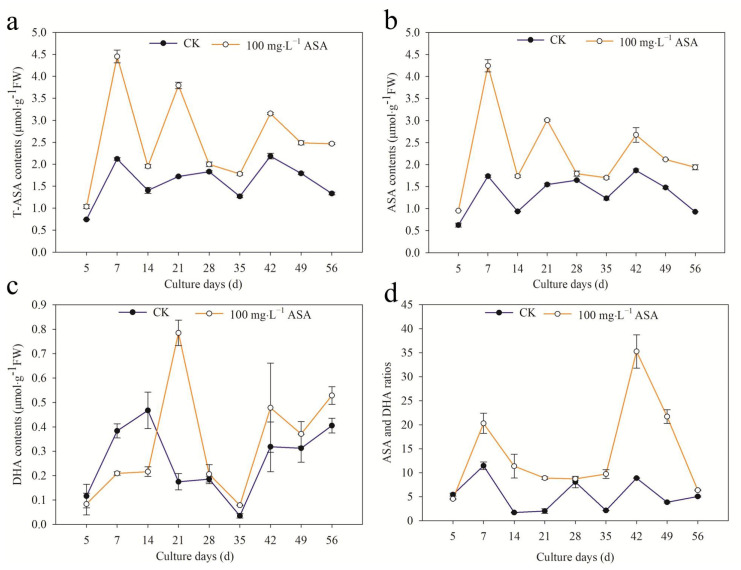
Effects of 0 and 100 mg·L^−1^ ASA treatments on ASA synthesis during somatic embryogenesis of *Fraxinus mandshurica* provenance 3. The *x*-axis represents the culturing days, the *y*-axis represents the T-ASA content (**a**), ASA content (**b**), DHA content (**c**), and ASA/DHA ratio (**d**), and the error bar shows the standard error of the mean.

**Figure 9 ijms-24-00289-f009:**
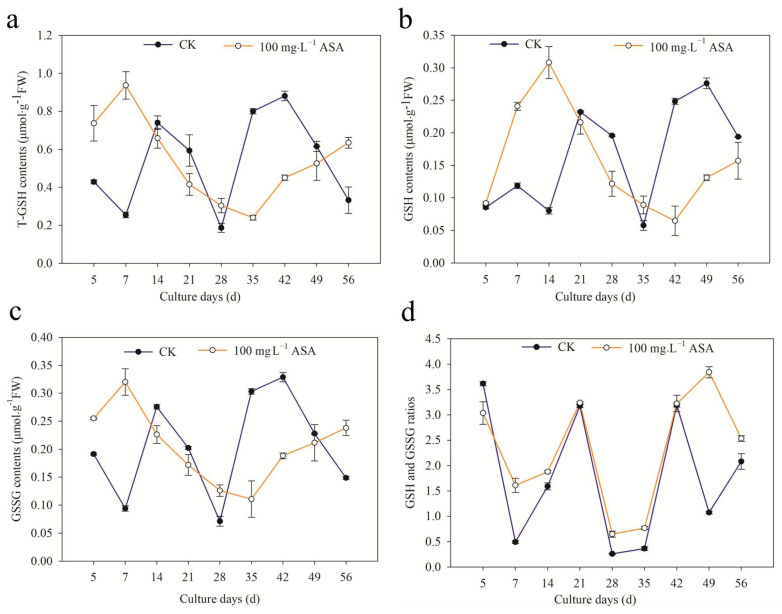
Effects of 0 and 100 mg·L^−1^ ASA treatments on GSH synthesis during somatic embryogenesis of *Fraxinus mandshurica* provenance 3. The *x*-axis represents the culturing days, the *y*-axis represents the T-GSH content (**a**), GSH content (**b**), GSSG content (**c**), and GSH/GSSG ratio (**d**), and the error bar shows the standard error of the mean.

**Figure 10 ijms-24-00289-f010:**
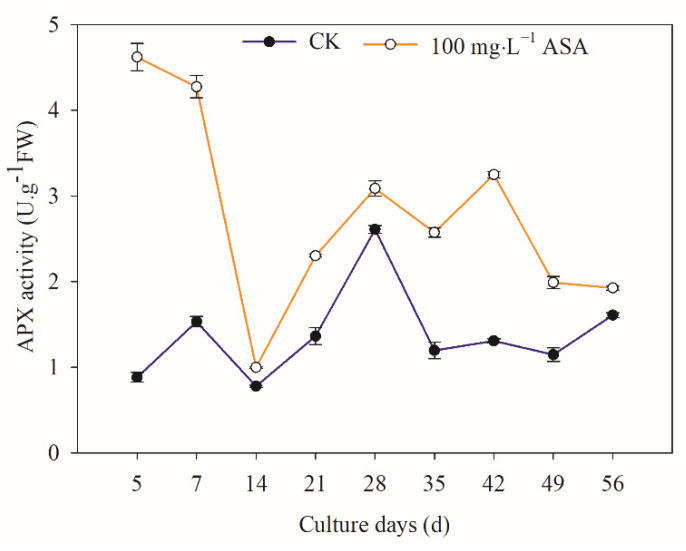
Effects of 0 and 100 mg·L^−1^ ASA treatments on APX activity during somatic embryogenesis of *Fraxinus mandshurica* provenance 3. The *x*-axis represents the culturing days, and the *y*-axis represents the APX activity. Error bars show the standard error of the mean.

## Data Availability

Not applicable.

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
