# Peer review of "Effects of Ascorbic Acid on Physiological Characteristics during Somatic Embryogenesis of Fraxinus mandshurica"

_ijms, 2022, doi:10.3390/ijms24010289_

Round 1

Reviewer 1 Report

The topic of manuscript is interesting. It discusses the impact of ascorbic acid (ASA) on somatic embryogenesis of Fraxinum mandshurica. However, some sections in the manuscript are very hard to comprehend. 

For example, 1) in line 100, the authors wrote "SEI rate in provenance 2 was 66.97%" (for ASA treatment), however the relevant Figure 2. indicates the provenance rate between 30 to 40%.

2) In line 115, authors state that PPO activity of day 35-42 d of ASA treatment was higher than control however in the relevant graph 3b. it appears to be lower than the control.

3) In line 141, authors mention that SOD and POD activity is obviously increased under ASA treatment, however, in the figure 4b the SOD activity of ASA treatment is lower than the control. 

4) In line 167-168, the authors mention "the content of H2O2 in the treatment group was 18.93%, which was higher than that in CK", this sentence doesn't match with the figure 5 as H2O2 concentration not percentage is listed. 

It is recommended that the authors should provide pictures of the explants comparing somatic embryogenesis and explant browning with and without ASA treatment(s)

Author Response

Dear Reviewer,

Our sincere thanks to you for the time and effort that you have put into reviewing our manuscript! We found all the comments very constructive and helpful, and have revised our manuscript according to all comments. Please find, below, our point-by-point response to the comments raised.

Thank you for considering our revised manuscript!

Point 1: in line 100, the authors wrote "SEI rate in provenance 2 was 66.97%" (for ASA treatment), however the relevant Figure 2. indicates the provenance rate between 30 to 40%.

Response 1: We apologise for this error, which has been revised in the text.

Point 2: In line 115, authors state that PPO activity of day 35-42 d of ASA treatment was higher than control however in the relevant graph 3b. it appears to be lower than the control.

Response 2: Because we draw a line chart, may cause you trouble, we are sorry. The PPO activity of 100mg.L-1 ASA treatment at 35-42 d was higher than that of the control group is correct.

Point 3: In line 141, authors mention that SOD and POD activity is obviously increased under ASA treatment, however, in the figure 4b the SOD activity of ASA treatment is lower than the control.

Response 3: We apologise for this error, which has been revised in the text.

Point 4: In line 167-168, the authors mention "the content of H2O2 in the treatment group was 18.93%, which was higher than that in CK", this sentence doesn't match with the figure 5 as H2O2 concentration not percentage is listed.

Response 4: We apologize for this misrepresentation, which has been revised in the text.

Point 5: It is recommended that the authors should provide pictures of the explants comparing somatic embryogenesis and explant browning with and without ASA treatment(s)

Response 5: We have supplemented it. For details, see: Figure 1A.

Reviewer 2 Report

 Your manuscript entitled ‘Effects of Ascorbic Acid on Physiological Characteristics During Somatic Embryogenesis of Fraxinus mandshurica’ has focused on the improvement of method(s) for somatic tissue culture in Fraxinus mandshurica. Your manuscript contains more interesting information about the physiological changing of somatic tissues of Fraxinus mandshurica’ during induction of somatic embryogenesis. However, the scientific novelty of your manuscript requires clarification.

Major points:

 1, Introduction:

After the second paragraph, you have to describe the practical importance of SE in Fraxinus mandshurica, for example, breeding, micropropagation, in vitro selection, genetic tranformation or genom editing etc.

2, Results:

You have to demonstrate the process and important steps of SE with some photos.

Figure 2: You have to use histograms or separated dots to demonstrate the data of experiments because four individual plants were used in the experiment. In this situation, you can not apply continuous lines.  

Figures: Fig 2 shows the significant difference of treatments among the provenances (genotypes) of Fraxinus mandshurica. Did you observe any genotypic or phenotypic differences among the donor plants (genotypes)? The genotype can influence the efficiency of SE. Which genotype and how many genotypes were used in the experiments?

In the captions of each Figures, you have to mention the cc. of ASA and the name of tested genotype(s).

Did you change, refresh (2-4 weeks) the media during the experiments? os Did you used the same media from 1. to 56. day of the experiment?   

3, Discussion:

Page 11 line 393 I didn’t find the correlation analyses in the Results section.

4, Materials and methods:

page 11 line 404 You have to describe the phenotypic and genotypic differences and similarities of the donor plants (prevalence or genotype).

5, Conclusion:

You have to summarize the scientific novelty of your experiments, and enhance the future perspectives, practical application of these experiments.

Minor points

page 7, line 223-225. This sentence is not clear. You have to rephrase it.

page 9, line 279 “In the 5-42 d.” This is not a complete sentence.

page 10, line 321 You should write “ASA. This…” instead of “ASA, This…”

page 10 line 326 You have to delete the dot “.” after ASA

page 10 line 359 You have to use italics in case of scientific name of plant species, Oryza sativa

page 12 line 423 I suggest using the “induction medium” instead of “inducement culture medium”.

page 12 line 424 My suggestion: Induction of somatic embryogenesis

page 12 line 427 Please correct the pH (instead of Ph).

Author Response

Dear Reviewer,

Our sincere thanks to you for the time and effort that you have put into reviewing our manuscript! We found all the comments very constructive and helpful, and have revised our manuscript according to all comments. Please find, below, our point-by-point response to the comments raised.

Thank you for considering our revised manuscript!

Point 1: Introduction: After the second paragraph, you have to describe the practical importance of SE in Fraxinus mandshurica, for example, breeding, micropropagation, in vitro selection, genetic tranformation or genom editing etc.

Response 1: We have supplemented the practical importance of SE in F. mandshurica.

Point 2: Results: You have to demonstrate the process and important steps of SE with some photos.

Response 2: We have supplemented the SE process and important steps, See for details: Figure 1A.

Point 3: Figure 2: You have to use histograms or separated dots to demonstrate the data of experiments because four individual plants were used in the experiment. In this situation, you can not apply continuous lines.

Response 3: We have redrawn the drawing with a histogram. See for details: Figure 2.

Point 4: Figures: Fig 2 shows the significant difference of treatments among the provenances (genotypes) of Fraxinus mandshurica. Did you observe any genotypic or phenotypic differences among the donor plants (genotypes)? The genotype can influence the efficiency of SE. Which genotype and how many genotypes were used in the experiments?

Response 4: We have supplemented the phenotypic differences of donor plant genotypes. See for details: Figure 1A. According to the comprehensive situation, provenance 3 was used for subsequent experiments.

Point 5: In the captions of each Figures, you have to mention the cc. of ASA and the name of tested genotype(s).

Response 5: We have added ASA concentration and test genotype name to the captions of each Figures.

Point 6: Did you change, refresh (2-4 weeks) the media during the experiments? or Did you used the same media from 1. to 56. day of the experiment?

Response 6: In the article, we have added the corresponding content.

Point 7: Discussion: Page 11 line 393 I didn’t find the correlation analyses in the Results section.

Response 7: We have rephrased these sentences accordingly.

Point 8:Materials and methods: page 11 line 404 You have to describe the phenotypic and genotypic differences and similarities of the donor plants (prevalence or genotype).

Response 8:It has been revised accordingly.

Point 9:Conclusion: You have to summarize the scientific novelty of your experiments, and enhance the future perspectives, practical application of these experiments.

Response 9:We have supplemented the content in the conclusion part.

Point 10:page 7, line 223-225. This sentence is not clear. You have to rephrase it.

Response 10:We have rewritten the sentence accordingly.

Point 11:page 9, line 279 “In the 5-42 d.” This is not a complete sentence.

Response 11:It has been revised accordingly.

Point 12:page 10, line 321 You should write “ASA. This…” instead of “ASA, This…”

Response 12: We have changed ASA, This to ASA. This.

Point 13:page 10 line 326 You have to delete the dot “.” after ASA

Response 13: We have deleted the dot “.” after ASA

Point 14:page 10 line 359 You have to use italics in case of scientific name of plant species, Oryza sativa

Response 14: Corrections have been made for scientific names.

Point 15:page 12 line 423 I suggest using the “induction medium” instead of “inducement culture medium”.

Response 15: We have made changes in the text.

Point 16:page 12 line 424 My suggestion: Induction of somatic embryogenesis

Response 16: We have made changes in the text.

Point 17:page 12 line 427 Please correct the pH (instead of Ph).

Response 17: We have corrected the PH.

Reviewer 3 Report

1. There are no pictures in the article, which makes it very difficult to understand what happened to the explants and in what form they were selected for analysis.

2. On what day was embryogenesis and browning of embryoids evaluated?

3. On which of the 4 populations were further measurements of e.g. peroxidase (POD) activity and others performed?

4. What was the sample for the analysis of e.g. peroxidase (POD) activity at day 42? How many explants, the explant was taken in its entirety, together with all the formed embryoids?

5. The final date of explant selection is day 56. Was the culture incubated on the same medium during all this time? Without reseeding? Explants must be crossed every 30 days or they will start to die.

6. The choice of ASA concentration is not certain. It is unacceptable to say that at 100 mg embryogenesis was maximal, if the next concentration that was tested caused complete death of the explants, connecting the points 100 and 150 on the graph is impossible, because obviously between them there could be at least 2 scenarios of events: 1 a sharp decrease, 2 an increase, then a decrease.

Author Response

Dear Reviewer,

Our sincere thanks to you for the time and effort that you have put into reviewing our manuscript! We found all the comments very constructive and helpful, and have revised our manuscript according to all comments. Please find, below, our point-by-point response to the comments raised.

Thank you for considering our revised manuscript!

Point 1: There are no pictures in the article, which makes it very difficult to understand what happened to the explants and in what form they were selected for analysis.

Response 1: We have supplemented the content of morphological changes during somatic embryogenesis of Fraxinus mandshurica. Please see details: Figure 1A.

Point 2: On what day was embryogenesis and browning of embryoids evaluated?

Response 2: We evaluated somatic embryogenesis and explants browning on the 56th day of somatic embryo induction culture. It has been supplemented in the article.

Point 3: On which of the 4 populations were further measurements of e. g. peroxidase (POD) activity and others performed?

Response 3: Combined with practical comprehensive analysis, provenance 3 was used for later physiological and biochemical analysis, which has been supplement in the text.

Point 4: What was the sample for the analysis of e. g. peroxidase (POD) activity at day 42? How many explants, the explant was taken in its entirety, together with all the formed embryoids?

Response 4: The sample used for determining POD activity on day 42 was the cotyledon explants grown on the medium for 42 days and the embryoids produced from them.

Point 5: The final date of explant selection is day 56. Was the culture incubated on the same medium during all this time? Without reseeding? Explants must be crossed every 30 days or they will start to die.

Response 5: We have updated the medium of explants every 30 days, which has been supplemented in this paper.

Point 6: The choice of ASA concentration is not certain. It is unacceptable to say that at 100 mg embryogenesis was maximal, if the next concentration that was tested caused complete death of the explants, connecting the points 100 and 150 on the graph is impossible, because obviously between them there could be at least 2 scenarios of events: 1 a sharp decrease, 2 an increase, then a decrease.

Response 6: When ASA concentration was 150 mg·L-1, the explants died, the browning rate and the somatic embryo rate were 0. Therefore, there was a sharp decline from ASA 100 mg·L-1 to 150 mg·L-1. See details: Figure 1B.

Round 2

Reviewer 1 Report

The authors have made significant improvements in the manuscript after the suggestion  provided during the last review. The revised manuscript is acceptable in its current form. I would still recommend the authors to go over the manuscript and make some minor spelling/grammar edits. 

Author Response

Dear Reviewer,

We are very grateful to you for taking the time to read and modify our article again. We find that your comments play a very important role in improving the quality of our papers. We have carefully revised the paper in light of your comments, and please find our response to the comments made below.

Thank you for considering our revised manuscript!

Point 1: The authors have made significant improvements in the manuscript after the suggestion provided during the last review. The revised manuscript is acceptable in its current form. I would still recommend the authors to go over the manuscript and make some minor spelling/grammar edits.

Response 1: Thank you very much for your recognition. We read the manuscript carefully and revised its grammar and spelling. For details please see the text.

Reviewer 2 Report

The authors answered my questions. The manuscript can be accepted. A grammar check can be useful.

Author Response

Dear Reviewer,

We are very grateful to you for taking the time to read and modify our article again. We find that your comments play a very important role in improving the quality of our papers. We have carefully revised the paper in light of your comments, and please find our response to the comments made below.

Thank you for considering our revised manuscript!

Point 1: The authors answered my questions. The manuscript can be accepted. A grammar check can be useful.

Response 1: Thank you very much for your recognition. We read the manuscript carefully and revised its grammar and spelling. For details please see the text.

Reviewer 3 Report

In Figure 1 on the graph of the browning of the embryoid, the line connecting 100 mg and 150 mg ASA cannot be drawn. With this line you guarantee that for example at concentration 120 mg ASA there was a suppression of embryoid formation. Are you sure about that? Maybe the stimulating effect lasted up to 120 mg or 130 mg and then began to decrease? At 100 mg you have marked a peak and to claim that further embryogenic activity and browning went down you need to give at least one point between the maximum and zero. If the explants died, how do you estimate their browning? My recommendation: If you have no data on the effect of ASA at concentrations between 100 and 150 mg/l, then remove the 150 mg point from the graph. Leave in the text the description that at this concentration the explants died, but do not show in the graph at all.

Author Response

Dear Reviewer,

We are very grateful to you for taking the time to read and modify our article again. We find that your comments play a very important role in improving the quality of our papers. We have carefully revised the paper in light of your comments, and please find our response to the comments made below.

Thank you for considering our revised manuscript!

Point 1: In Figure 1 on the graph of the browning of the embryoid, the line connecting 100 mg and 150 mg ASA cannot be drawn. With this line you guarantee that for example at concentration 120 mg ASA there was a suppression of embryoid formation. Are you sure about that? Maybe the stimulating effect lasted up to 120 mg or 130 mg and then began to decrease? At 100 mg you have marked a peak and to claim that further embryogenic activity and browning went down you need to give at least one point between the maximum and zero. If the explants died, how do you estimate their browning? My recommendation: If you have no data on the effect of ASA at concentrations between 100 and 150 mg/l, then remove the 150 mg point from the graph. Leave in the text the description that at this concentration the explants died, but do not show in the graph at all.

Response 1: Thank you again for your opinion, we carefully studied your suggestion, and agreed that what you said is very reasonable. We have modified Figure 1 according to your suggestion, and supplemented the explant state with ASA concentration of 150 mg∙L-1 in the text. Your suggestions give us a lot of experimental inspiration, which is of great significance to improving our paper. We put the explant death, its browning rate and somatic embryo rate recorded as 0, maybe some inappropriate, we must pay attention to next time.